# Preparation and Characterization of Carbide Particle-Toughened Si–B System of High Thermostability Polycrystalline Diamond by HPHT Sintering

**DOI:** 10.3390/ma16113933

**Published:** 2023-05-24

**Authors:** Yunqi Zhang, Yumei Zhu, Zhihong Li

**Affiliations:** Key Laboratory for Advanced Ceramics and Machining Technology of Ministry of Education, School of Materials Science and Engineering, Tianjin University, Tianjin 300072, China; zyq1998@tju.edu.cn (Y.Z.); zhuyumei@tju.edu.cn (Y.Z.)

**Keywords:** PCD composites, the second phase, thermal properties, mechanical properties

## Abstract

In this research, we report the synthesis of Si–TmC–B/PCD composites using Si, B, and transition metal carbide particles (TmC) as binders at high pressure and high temperature (HPHT method, 5.5 GPa and 1450 °C). The microstructure, elemental distribution, phase composition, thermal stability, and mechanical properties of PCD composites were systematically investigated. The Si–B/PCD sample is thermally stable in air at 919 °C. The initial oxidation temperature of the PCD sample with ZrC particles is as high as 976 °C, and it also has a maximum flexural strength of 762.2 MPa, and the highest fracture toughness of 8.0 MPa·m^1/2^.

## 1. Introduction

The hardest substance known to man is diamond, and natural diamond can reach a Vickers hardness of 70–120 GPa [1]. It also offers excellent properties such as high wear resistance, high strength, high thermal conductivity, low coefficient of thermal expansion, and good biocompatibility [2,3,4,5]. Since the successful synthesis of manmade diamonds, the polycrystalline diamond (PCD) industry has grown rapidly [6]. It is widely used in oil drilling, aerospace, geological exploration, machining, and biomedicine [7,8,9,10,11,12]. However, when PCD drill bits are used for drilling in abrasive hard rock formations or nonuniform rock formations, their rock-breaking performance and drilling speed decrease rapidly, and the life of the drill bit is also limited by the thermal stability of PCD [13].

The major reason for the poor thermal stability of PCD is due to the existence of metal sintering aids such as Fe, Ni, Co, and Cr. Co is one of the most common metal sintering aids in the synthesis of PCD; however, Co also acts as a catalyst for the conversion of diamond to graphite [14]. Owing to the catalytic effect of Co, which leads to the reduction of its thermal stability, the graphitization temperature of PCD decreases to 700 °C, which seriously affects the lifetime of PCD at high-temperature conditions [15,16]. In addition, the substantial difference in the coefficient of thermal expansion between diamond (~1.2 × 10^−6^) and Co (~12.3 × 10^−6^) leads to thermal stresses and microcracks at high temperatures, which weakens the performance of PCD [17].

With the goal to enhance the thermal stability of PCD materials, the following methods are commonly used. Etching the metal by electrolysis or chemical treatment, however, by removing of the Co creates a lot of pores, which leads to a dramatic reduction in the strength and hardness of PCD [18,19]. The process of leaching also incurs additional costs, and the PCD substrate is more difficult to braze without a metal binder. Alternatively, alkaline earth carbonates such as MgCO_3_, Li_2_CO_3_, and CaCO_3_ are used instead of metals as sintering aid [20]. Westraadt et al. [13] concluded that the function of CaCO_3_ as a sintering aid is a catalytic solvent, comparable to the function of metals in conventional PCD, and no significant changes were found in the flank wear scars produced by processing silicone rods after vacuum heat treatment at 1200 °C in sintered PCD, indicating better wear resistance and thermal stability of PCD. Tian et al. [1] synthesized the first direct synthesis of nanotwinned diamond with an average twin thickness of 5 nm using nano-onion carbon precursors at 20 GPa and 2000 °C using no sintering aid, and due to the lower energy of the twin boundaries, the presence of ultrafine nanotwins retarded the graphitization and oxidation of diamond, and the oxidation temperature in air (~980 °C) was 210 °C higher than that of natural diamond (~770 °C). Another alternative is to use a chemically noble bonding phase, such as SiC, which is usually formed by reacting Si with diamond. As the melting point of Si (1400 °C) decreases with increasing pressure, it provides a stable temperature area for the diamond. The high melting point of the product, silicon carbide, ensures adaptability to high-temperature environments. Liu et al. [14] investigated a new triple-layer structure of polycrystalline diamond compact (PDC), in which WC/Co cemented carbide was used as the substrate, diamond–SiC–Co composite as the intermediate layer, and diamond–SiC composite as the working layer. The Co in the substrate was prevented from penetrating into the working layer, which enhanced the thermal stability, and it was shown that the initial oxidation temperature of the triple-layer structure PDC was increased by 40 °C compared to that of the conventional PDC (~780 °C). Si-coated diamond prepared by chemical vapor deposition effectively prevents the existence of oxygen-containing functional groups on the diamond surface, and Si-coated PCD has 12% and 30% higher onset oxidation temperature and wear resistance than uncoated PCD, respectively [21]. Contrary to Co/PCD products, this material typically has a poorer fracture toughness. [13].

In this paper, Si–TmC–B/PCD (TiC, ZrC, VC, NbC, and WC) materials with high thermal stability and high toughness were prepared by sintering under high pressure and high temperature conditions using Si–B/PCD as a control experiment (CE). Si and diamond generate SiC in situ, which improves the thermal stability of PCD materials, and small-size transition metal carbides and B are introduced to fill the internal voids and play the role of fine grain strengthening while generating excellent performance of transition metal diboride (TmDB), which improves the phase composition of the sintered body. Systematic studies on the microscopic morphology, phase component, and thermomechanical properties of the Si–TmC–B/PCD system were carried out.

## 2. Experimental Methods

### 2.1. Fabrication of Samples

Starting materials included diamond powder, silicon powder, boron powder, and TmC powder (TiC, ZrC, VC, NbC, and WC). The diamond powder (with an average particle size of 1 μm, 3.5 μm, and 10 μm and a purity of 99.99%) was supplied by Zhongnan Jete Superhard Materials Co., Ltd. (Zhengzhou, China). The silicon powder (with an average particle size of 1 μm and a purity of 99.99%), boron powder (with an average particle size of 0.8–5 μm and a purity of 99.5%), and TmC powder (with an average particle size of 1–2 μm and a purity of 99.9%) were supplied by Eno High-Tech Materials Development Co. Ltd. (Qinhuangdao, China). PCD consisted of a mixture of diamond powder and binder powder. According to the previous experiments, the mass ratio of diamond powder to binder powder was 85:15. The diamond powder and binder powder (Si:B = 95:5 for Si–B/PCD sample and Si:TmC:B = 60:35:5 for Si–TmC–B/PCD samples) were mixed in a planetary ball mill. Then we added anhydrous ethanol and ZrO_2_ grinding balls with a ball to powder ratio of 2:1 ratio into the ball mill and mixed the mixture thoroughly at 400 r/min for 3 h. After vacuum drying for 24 h, it was passed through a 120 mesh screen. In order to eliminate gas adsorption on the surface of the material, it was treated in an H_2_ atmosphere (pressure of 3.0 × 10^−3^ Pa) at 380 °C for 24 h. The sintering process was performed in a Hinge-Type Six-Anvil Press with sintering conditions of HPHT (5.5 GPa, 1450 °C) and a holding time of 90 s. The sintered samples were cylinders of 26 mm in diameter and 6 mm in thickness. They were cut by laser into 4 mm × 6 mm × 24 mm rectangular shapes and mirror-polished for testing and characterization.

### 2.2. Characterization of Samples

The microscopic morphologies of the samples were measured by SEM (S-4800, Hitachi, Tokyo, Japan). The elemental composition of different regions was determined by energy dispersive spectroscopy (EDS, X-MAX20, Oxford, UK). Phase composition was performed by X-ray diffraction (XRD, D/MAX-2500, Rigaku, Tokyo, Japan). The flexural strength of the samples was tested by a three-point bending experiment (XWW, Beijing Jinshengxin Detecting Instrument Co., Ltd., Beijing, China) with a loading rate of 0.5 mm/min. The density of the samples was determined by an electronic balance (BSA224S-CW, Sartorius Scientific Instruments Co., Ltd., Beijing, China). The fracture toughness was measured by using a Vickers diamond indenter hardness durometer (HMAS-010, Shanghai Runguang Technology Co., Ltd., Shanghai, China). The thermal stability of the samples was determined by thermogravimetry and differential scanning calorimetry in an air atmosphere using a simultaneous thermal analyzer (NETZSCH STA, 449F3, Selb, Germany) with a heating rate of 10 °C/min and a heating range of 40–1300 °C. The samples were prepared into CNMN 120408 type inserts for straight turning of Ti-6Al-4V (TC4) alloy on a CNC lathe (CK6136, Guangzhou Numerical Control Equipment Co., Ltd., Guangzhou, China) shown in Figure 1. The tool wear condition was measured under an optical digital microscope (MUSTCAM, Shenzhen Maike Vision Electronics Co., Ltd., Shenzhen, China), and the average flank wear VBa ≥ 0.3 mm was used as the tool failure criterion [22]. The summary of working parameters is shown in Table 1.

The three-point bending experiments were performed on rectangular samples [23], and Equation (1) was used to calculate the value of flexural strength.
(1)$$\delta =\frac{3FL}{2bh^{2}}$$
where *F* is the damage load, *L* is the span, and *b* and *h* are the width and thickness.

According to the Vickers microcracks indentation method (VIM), we evaluated the fracture toughness of the samples [12]. The fracture toughness value was calculated using Equation (2) with a trial load of 30 N.
(2)$$K_{IC}=0.016{\left(\frac{E}{H_{v}}\right)}^{\frac{1}{2}}\frac{F}{C^{\frac{3}{2}}}$$
where *E* is the Young’s modulus, *Hv* is the Vickers hardness, *F* is the trial load, and *c* is the average radial half-crack length measured from the center of the indentation. The Oliver and Pharr methods were used to obtain the Young’s modulus [24,25].

## 3. Results and Discussion

### 3.1. Microstructure and Phase Composition Analysis

Figure 2 displays the microstructure of PCD samples, both Si–B/PCD samples as a control experiment and Si–TmC–B/PCD samples; they both have only a few holes and no obvious cracks. The dark-colored blocky area is diamond particles, and the light-colored area is the binder, which is tightly wrapped around the diamond particles. According to the cross-grain fracture characteristics on the diamond fracture surface, the bond strength of the binder to the diamond is stronger than the diamond’s fracture strength in specific planes and orientations, which results from the rapid sintering procedure at high temperatures and pressure. Si fills in between diamond particles in a molten state and reacts with the diamond in situ, producing SiC; B facilitates the interfacial penetration of diamond and Si particles and also binds to the dangling bonds of the diamond surface, making a dense sintered body out of the diamond and binder [26].

Figure 3 displays the XRD pattern of the PCD samples. The main crystalline phases in the samples are diamonds and SiC, and the Si–B/PCD samples as a control experiment have relatively high Si powder content in the initial powder, due to the nonaddition of carbide particles, and thus contain residual Si, and the Si in the rest of the samples has been completely consumed. In other samples, in addition to transition metal carbides, a small amount of transition metal diborides which have high hardness, strong wear resistance, and high temperature resistance properties are detected [27]. It indicates that the sintering drive is sufficient at 1450 °C and 5.5 GPa, and the homogeneous and dense PCD composites are successfully prepared.

The Si–ZrC–B/PCD samples with high thermal stability and excellent mechanical properties were analyzed by energy dispersive spectroscopy (Figure 4). In Figure 4c–f, it is easy to see that Zr and Si are mainly distributed at the grain boundaries, and B is diffusely distributed in the sample, which is due to the similar atomic radius of both boron and carbon atoms, which can easily undergo position substitution and enter the internal diamond interstices. It plays the role of filling internal vacancies and other defects and improving the internal structure. The line scanning at the grain boundaries shows that the peaks of C, Zr, and B elements, as well as C and Si elements, appear to increase significantly at the grain boundaries, indicating that the three substances, SiC and the in situ-generated ZrB_2_ and ZrC, act together at the grain boundaries to form a dense sintered body with the diamond grains.

### 3.2. Thermal Stability Analysis

The TG–DSC method is used to characterize the thermal stability of PCD samples in an air environment. The top of the exothermic peak cannot accurately represent the thermal stability due to the degradation of the PCD mechanical properties caused by the strong chemical reaction; therefore, it is reasonable to adopt the initial oxidation temperature as the evaluation criterion for the thermal stability of PCD samples.

As shown in Figure 5, the thermal stability of the Si–TmC–B/PCD samples was universally higher than that of the Si–B/PCD sample, and the initial oxidation temperature increased by 33–57 °C. The sample with the additive ZrC particles has the highest thermal stability (~976 °C) (Figure 5c), and the Si–TiC–B/PCD sample has the least enhancement of thermal stability (~952 °C) (Figure 5b). Si is more active than SiC and diamond in the presence of air, and, as a result, the oxidation of Si remaining in the area of the binder in Si–B/PCD sample occurs first, and the increase in temperature and the exothermic process of Si oxidation accelerates the oxidation reaction of SiC and diamond [21]. With the addition of TmC particles, the relative content of silicon in the raw material decreases, and all silicon at the grain boundaries reacts to form SiC with a high melting point and excellent antioxidant properties. SiC prevents oxygen from coming into contact with diamond and slows down the oxidation of the diamond. Meanwhile, TmC particles have excellent properties of high melting point, hardness, and thermal conductivity, and the introduction of TmC particles is beneficial to enhance the thermomechanical properties of PCD [28]. The coefficient of thermal expansion of the introduced TmC particles has a minimum value of 6.22 × 10^−6^ K^−1^ for ZrC and a maximum value of 7.74 × 10^−6^ K^−1^ for TiC [29,30]. During the heating up and oxidation process, the Si–ZrC–B/PCD sample generates the least thermal stress inside the material; therefore, this sample has the highest thermal stability. In addition, the uniformly distributed boron atoms cause the excess valence electrons on the diamond surface to be bound, eliminating the hanging bonds on the diamond surface so that there are no excess valence electrons to react with the oxygen atoms. Boron atoms react with oxygen when heated to form a low-melting-point B_2_O_3_, which reacts with metal oxides in the molten state to form stable borates, thus forming a protective film on the diamond surface, reducing the rate of oxidation and improving the thermal stability of PCD. The thermal stability of Si–ZrC–B/PCD samples increases by 235 °C and 262 °C compared to other Si/PCD and Co/PCD [21,26].

TC4 alloy, which has high strength and corrosion resistance, generates high cutting temperatures during machining due to its high chemical reactivity and low thermal conductivity [9,31]. Using the PCD tool for straight turning of TC4 alloy, the variation of average flank wear VBa with cutting length L of the PCD tool is shown in Figure 6. It can be found that when VBa = 0.3 mm and tool failures occur, PCD with TmC particles has a general increase in cutting length compared to Si–B/PCD, increasing up to 26.2–120.2%. The Si–ZrC–B/PCD sample with the longest cutting length shows the highest wear resistance with a life of 5162 m. It is easily seen that the PCD tool cutting length variation law is basically consistent with the results of the thermal stability of PCD specimens in Figure 5; the higher the thermal stability, the larger the cutting length of the tool. This is because when the tool is close to failure, the friction generated by the tip dulling causes a rapid accumulation of heat and a sharp increase in tool wear, which eventually leads to tool failure.

### 3.3. Mechanical Properties Analysis

Figure 7 shows the trend of flexural strength and relative density of the samples. Generally speaking, the density of PCD is extremely high under HPHT synthesis conditions, and the relative density of the Si–B/PCD sample can reach 97.5%. After adding TmC particles, the relative density of PCD increases significantly, where the Si–ZrC–B/PCD sample has the maximum flexural strength of 762.2 MPa while obtaining the maximum relative density of 99.72%, and the flexural strength increases up to 14.1%. It is well known that the generation of vacancies is closely related to the ionic radius, and the larger the radius mismatch between cations, the higher the number of vacancies generated and the lower the activation energy for crystal growth [32]. The ionic radius difference between Zr^4+^ and Si^4+^ is the largest, r(Zr^4+^)/r(Si^4+^) = 1.8, so the introduction of ZrC has a strong promotion effect on the sintering of Si–B system PCD, which makes Si–ZrC–B/PCD obtain excellent mechanical properties. In the figure, it can be seen that the trend of flexural strength tends to be consistent with the change in relative density. Normally, the factors that affect the strength of a material include composition, porosity, and grain size. In the control experiment, the residual unreacted Si causes a decrease in the bond strength at the diamond grain boundaries; therefore, this sample has a lower ability to resist bending without fracture. In contrast, in the Si–TmC–B/PCD samples, the relative content of Si in the raw material decreased due to the introduction of TmC particles with excellent mechanical properties, and dense sintered bodies are synthesized under HPHT conditions; in addition, fine diffuse TmC particles (Figure 4) are uniformly distributed in the matrix as a second phase, producing a significant dispersion strengthening effect, which is advantageous to enhancing the mechanical qualities of the material [33].

The fracture toughness of PCD samples was measured using the indentation method, and the results are shown in Figure 8. The fracture toughness of TmC particle samples was improved by 25–45%, and the toughness of ZrC particle samples was as high as 8.0 MPa·m^1/2^. Figure 9 shows the high-magnification FESEM image. In Figure 9a, obvious cracks and fracture features of intercrystalline are visible, which indicates that the sample is easily fractured from the grain boundaries after being pressured because of the low bonding strength at the grain boundaries. The TmC particles are introduced into the PCD matrix as a second phase, which plays the role of fine grain reinforcement. In addition, the transition metal diboride generated by the in situ reaction with the diffusely distributed B atoms is uniformly distributed at the grain boundaries, and the degree of densification is further enhanced. The typical feature of river stripes in Figure 9b–f indicates that the bonding strength at the diamond grain boundaries is enhanced above the fracture strength in the specific direction and plane of the diamond; thus, transcrystalline fracture occurs. The variation of fracture pattern from intercrystalline fracture to mixed fracture dominated by transcrystalline fracture is a typical feature of increased fracture toughness [34]. TmC particles also hinder further crack expansion by pinning the crack at the crack tip [35]. In addition, the mismatch of the elastic modulus between the TmC particle and the binder leads to a load transfer effect that consumes the energy required for crack expansion and therefore provides a toughening effect [36].

## 4. Conclusions

In this experiment, Si–TmC–B/PCD composites were synthesized using Si, B, and transition metal carbide particles as binders at 5.5 GPa and 1450 °C. The results show that the thermal stability of PCD is significantly improved. With the addition of carbide particles, the flexural strength and fracture toughness of PCD samples are greatly enhanced.

1. No large holes and cracks are observed in the microstructure of the prepared PCD. The generation of SiC and a small amount of transition metal diboride indicates that the sintering driving force is sufficient and the components within the PCD are tightly bonded and form a dense sintered bulk.

2. The transition metal carbide, SiC, and transition metal diboride are tightly bonded at the grain boundaries, effectively preventing oxygen contact with the diamond and slowing down the oxidation of PCD. The initial oxidation temperature of the Si–ZrC–B/PCD sample is up to 976 °C.

3. The transition metal carbide particle, as a second phase, plays a significant role in fine grain strengthening, which results in excellent flexural strength of Si–ZrC–B/PCD sample, up to 762.2 MPa.

4. The transition metal carbide particle changes the fracture mode of PCD, forming a mixed fracture mode dominated by transcrystalline fracture, and also plays the role in pinning the crack and hindering the crack expansion. In addition, the mismatch of the elastic modulus between the transition metal carbide particle and the binder causes a load transfer effect that consumes the energy required for crack extension.

5. Future studies can explore the effect of nanoscale transition metal carbide particles on the thermal stability and mechanical properties of PCD.

## Figures and Tables

**Figure 1 materials-16-03933-f001:**
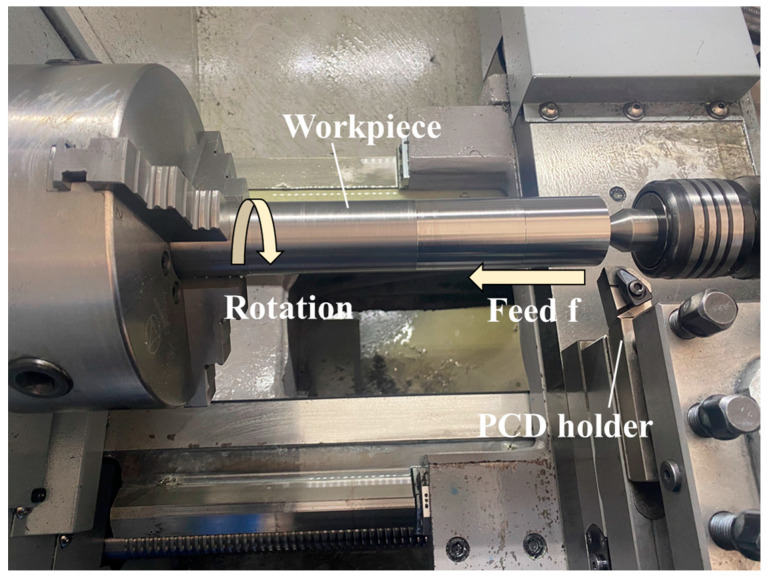
Machine tool setup.

**Figure 2 materials-16-03933-f002:**
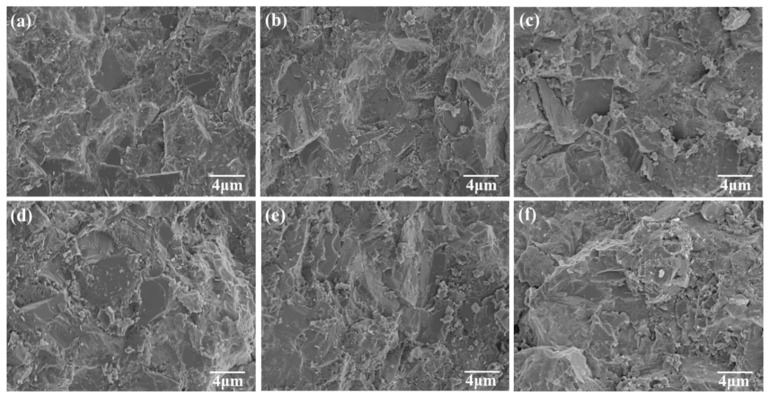
FESEM images of Si–TmC–B/PCD samples, (**a**) Si–B/PCD, (**b**) Si–TiC–B/PCD, (**c**) Si–ZrC–B/PCD, (**d**) Si–VC–B/PCD, (**e**) Si–NbC–B/PCD, and (**f**) Si–WC–B/PCD.

**Figure 3 materials-16-03933-f003:**
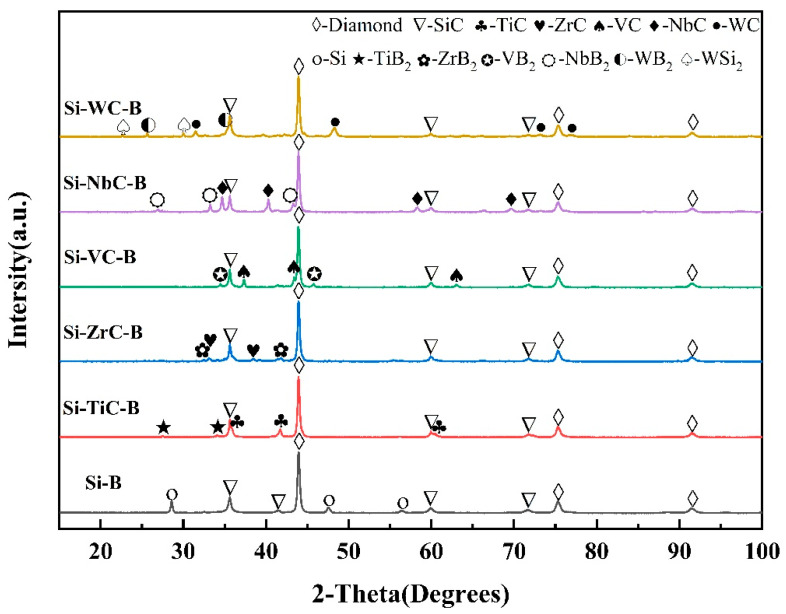
XRD of PCD samples with different additives.

**Figure 4 materials-16-03933-f004:**
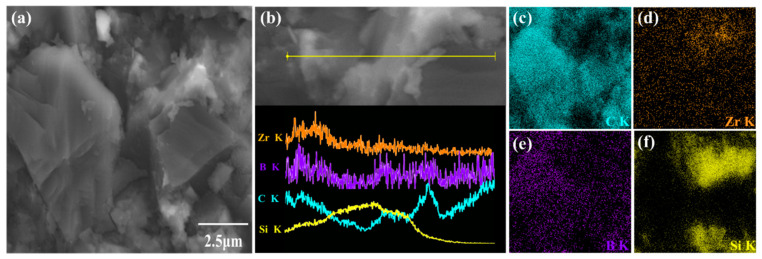
FESEM image (**a**), line-scanning (**b**) of fracture Si–ZrC–B/PCD sample, and corresponding EDS maps of (**c**) C, (**d**) Zr, (**e**) B, and (**f**) Si.

**Figure 5 materials-16-03933-f005:**
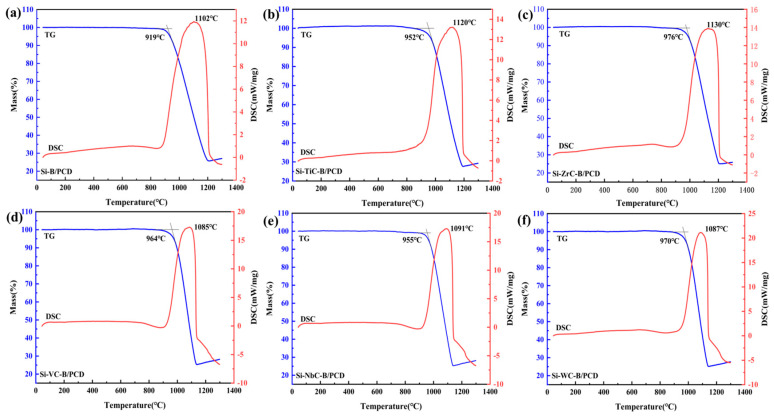
TG–DSC patterns of the PCD samples, (**a**) Si–B/PCD, (**b**) Si–TiC–B/PCD, (**c**) Si–ZrC–B/PCD, (**d**) Si–VC–B/PCD, (**e**) Si–NbC–B/PCD, and (**f**) Si–WC–B/PCD.

**Figure 6 materials-16-03933-f006:**
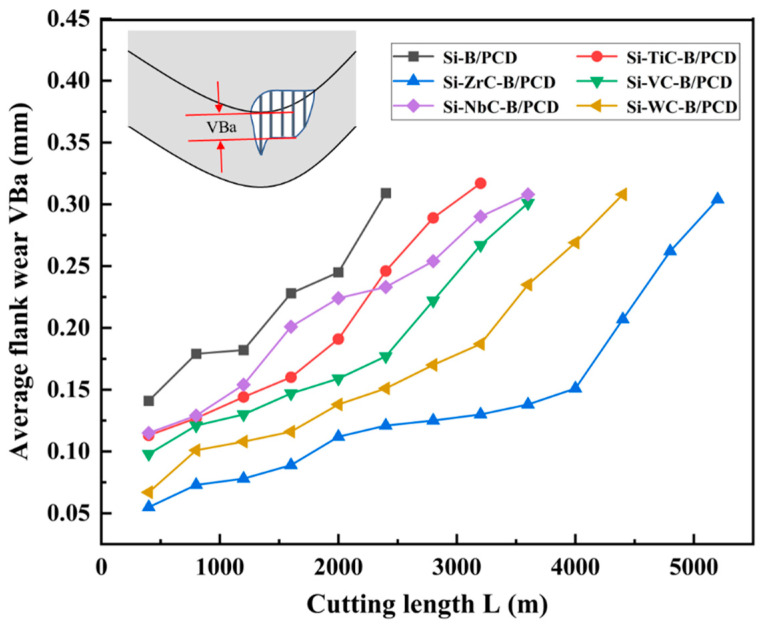
Progressive flank wear at V = 100 m/min, f = 0.15 mm/r, and α = 0.1 mm.

**Figure 7 materials-16-03933-f007:**
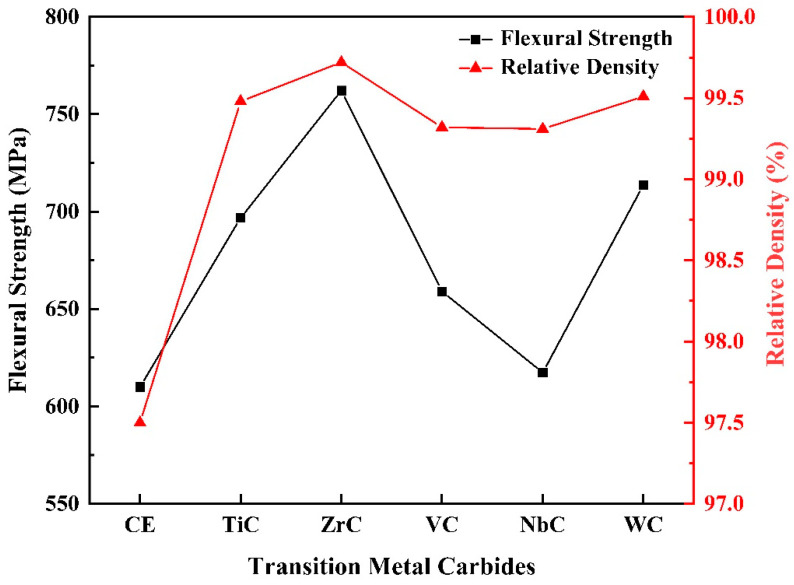
The flexural strength and relative density of Si–B/PCD and Si–TmC–B/PCD samples.

**Figure 8 materials-16-03933-f008:**
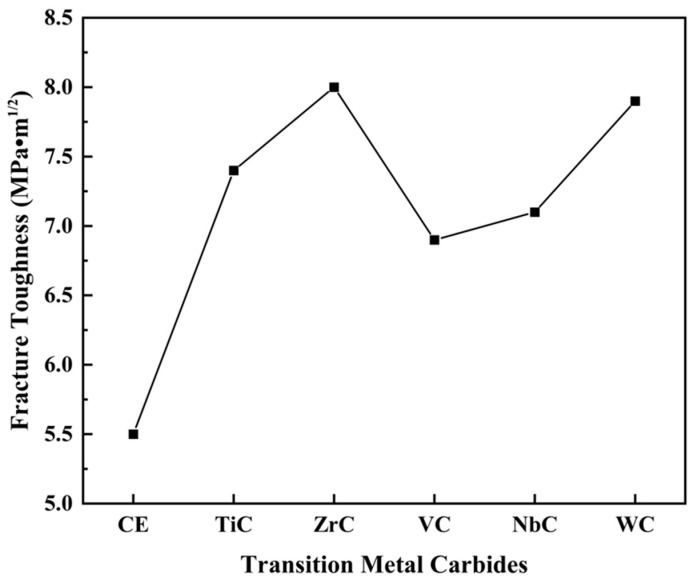
Variations trend in fracture toughness of Si–B/PCD and Si–TmC–B/PCD samples.

**Figure 9 materials-16-03933-f009:**
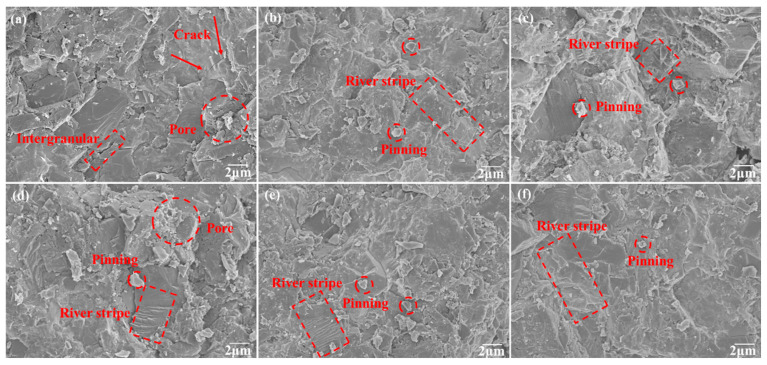
FESEM images of the toughening process, (**a**) Si–B/PCD, (**b**) Si–TiC–B/PCD, (**c**) Si–ZrC–B/PCD, (**d**) Si–VC–B/PCD, (**e**) Si–NbC–B/PCD, and (**f**) Si–WC–B/PCD.

**Table 1 materials-16-03933-t001:** Experimental conditions for turning.

Cutting Tool		PCD
Rake angle	α	0°
Nose radius	r_ε_	0.8 mm
Processing material		TC4
Cutting velocity	v	100 m/min
Cutting depth	α	0.1 mm
Feed rate	f	0.15 mm/rev
Cutting method		Dry
Air supply		0.7 MPa

## Data Availability

Data are contained in the article.

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
