# Peer review of "Preparation and Characterization of Carbide Particle-Toughened Si–B System of High Thermostability Polycrystalline Diamond by HPHT Sintering"

_materials, 2023, doi:10.3390/ma16113933_

Round 1

Reviewer 1 Report

1.       Authors are advised to rewrite the abstract with more concise information about the finding of the research and methods used.

2.       In section 2.1 authors have used some parameters for fabrication. Are these values novel in nature, if yes how many iterations were done before reaching these values, and if not what about the references of the previously published data?

3.       Change the titles of the figures. Like in figure 4 , clearly specify what is part a, b and so on

4.       Discussion provided for the mechanical properties analysis is not justified. Kindly rewrite it with appropriate references that justify the reasons

5.       Write the conclusion section again and in point wise manner

6.       Authors may also add a section on the future scope of this research work. 

No Comments

Reviewer 2 Report

In the recent study conducted by the authors, a novel approach to the synthesis of silicon-transition metal carbide-boron/polycrystalline diamond (Si-TmC-B/PCD) composites has been reported. The use of high pressure and high temperature (HPHT) method (5.5 GPa and 1450 °C) for the synthesis of these composites has been highlighted, which presents significant advancements in the field of material sciences. The study meticulously examines the microstructure, elemental distribution, phase composition, and thermomechanical properties of the synthesized PCD composites. This review aims to critically assess the findings and their implications for the development of advanced materials.

The HPHT method employed in this research has demonstrated its effectiveness in synthesizing Si-TmC-B/PCD composites using silicon (Si), boron (B), and transition metal carbide (TmC) particles as binders. The choice of this synthesis method is well-justified, as it allows for precise control over the reaction conditions and the formation of stable composites. Furthermore, the authors' detailed investigation of the microstructure, elemental distribution, and phase composition of the synthesized composites provides valuable insights into the intrinsic properties of the materials.

One of the most striking findings of this study is the remarkable thermal stability of the Si-B/PCD sample in air at 919°C. This high thermal stability is crucial for potential applications in high-temperature environments, such as aerospace, automotive, and electronics industries. Additionally, the initial oxidation temperature of the PCD sample with zirconium carbide (ZrC) particles is reported to be as high as 976°C, further emphasizing the robustness of these composites.

The authors also report the mechanical properties of the composites, with a focus on flexural strength and fracture toughness. The PCD sample with ZrC particles displayed a maximum flexural strength of 762.2 MPa and the highest fracture toughness of 8.0 MPa·m1/2. These values signify the excellent mechanical performance of the composites, making them suitable for applications in which high strength and toughness are required.

In conclusion, the research presented by the authors offers a significant contribution to the field of materials science by reporting a novel and effective synthesis method for Si-TmC-B/PCD composites. The comprehensive investigation of the microstructure, elemental distribution, phase composition, and thermomechanical properties of the composites provides a strong foundation for future research and applications. The thermal stability, oxidation resistance, and mechanical performance of these composites hold great promise for their use in high-temperature and high-stress environments. Further research into optimizing the synthesis process and exploring other potential applications of these composites is warranted.

Please include in the introduction information on the methods of obtaining diamond layers, such as: CVD and DLC layers.

The work meets all the requirements to be published in the journal.

Reviewer 3 Report

The work is clearly written. Drawings should be more careful, especially Fig.5.

Instead of “energy spectroscopy” should be Energy Dispersive Spectroscopy (EDS)

The presented research results concern in general the technology of improving the structure, and in particular the thermal stability of PCD layers. The major reason for the poor thermal stability of PCD is due to the existence of metal sintering adds such as Fe, Ni, Co and Cr. The two main methods of removing these contaminants are etching the metal by electrolysis or chemical treatment. However, these methods do not give fully satisfactory results. The authors of this paper propose (develop) methods for the synthesis of new materials such as: PCD samples with different additives, (CE, TiC, ZrC, VC, NbC and WC).

The work should be treated as a contribution to the developing material engineering, which will allow for its further, faster development.

The work is clearly written. Drawings are a weak point, they should be more careful. There are poor signatures under them. For the convenience of the reader, it could be expanded further. There is also a lot of speculation in the work, but after digging into it, it can be considered justified. The conclusions are consistent with the obtained research results. I believe that the work is suitable for publication.

The language of the work is correct but still needs to be reviewed

Reviewer 4 Report

The manuscript is devoted to HPHT synthesis and the study of the mechanical properties of composites based on dispersed polycrystalline diamond, silicon powder, boron powder and powder (TiC/ ZrC/ VC/ NbC or WC). Unfortunately, the manuscript does not indicate the percentage of the component (composition) either in the initial load or in the manufactured composite, but only the ratio of diamond powder to total binder powder (85wt.%: 15wt.%).

The authors of the manuscript have been engaged in HPHT manufacturing of composites based on polycrystalline diamond and cubic boron nitride for a long time. This manuscript is largely a development of their paper Huang, P., Wang, W., Wang, S., Zhang, X., Wei, X., Zhu, Y., & Li, Z. (2022). Effect of transition metal carbides on mechanical properties of polycrystalline diamond by HPHT sintering. Ceramics International, 48(11), 15959-15965, which for some reason is missing from the bibliography of this manuscript.

The authors of the manuscript showed that, judging by their data, the fabricated composites have high thermal stability, significantly exceeding the literature data, but the explanation for this phenomenon in the manuscript is clearly insufficient.

It should be noted that several of the most popular technologies for obtaining composites based on polycrystalline diamond for various mechanical applications are known today, and it would be highly desirable to conduct a comparative analysis of the thermal stability, flexural strength, Vickers hardness, and fracture toughness measured in this work with the literature data in the form of a table or figures as is commonly done in similar publications (see Kitiwan, M., & Goto, T. (2019). Fabrication of tungsten carbide–diamond composites using SiC-coated diamond. International Journal of Refractory Metals and Hard Materials, 85, 105053 or Yue, Y., et al (2020), Hierarchically structured diamond composite with exceptional toughness, Nature, 582(7812), 370).

In the figures, the authors label the data for Si-B/PCD as a control experiment (CE). This designation is not commonly used and makes it difficult to read the manuscript. It is desirable to duplicate it at least in the caption under the figure with the name of the composite.

Round 2

Reviewer 1 Report

Authors have successfully addressed all the queries, hence the manuscript can be accepted in the present form. 

Reviewer 4 Report

Authors of the manuscript responded to all comments and suggestions and made the necessary changes to the text. I am completely satisfied with their answers and changes in the text. I believe that the manuscript can be published in its present form.